# Satellite-Based Drought Impact Assessment on Rice Yield in Thailand with SIMRIW–RS

**Mongkol Raksapatcharawong [1,*] and Watcharee Veerakachen [1], Koki Homma [2], Masayasu Maki [3] and Kazuo Oki [4,5]**

[1] Chulabhorn Satellite Receiving Station, Faculty of Engineering, Kasetsart University, Bangkok 10900, Thailand; watcharee.v@ku.th

[2] Graduate School of Agricultural Science, Tohoku University, Sendai 981–8555, Japan; koki.homma.d6@tohoku.ac.jp

[3] Faculty of Food and Agricultural Sciences, Fukushima University, Fukushima 960–1298, Japan; makimasa@agri.fukushima–u.ac.jp

[4] Institute of Industrial Science, The University of Tokyo, Tokyo 153–8505, Japan; kazu@iis.u–tokyo.ac.jp or oki.kazuo@kuas.ac.jp

[5] Faculty of Engineering, Kyoto University of Advanced Science, Kyoto 615–8577, Japan

[*] Correspondence: mongkol.r@ku.th; Tel.: +66–81–694–1188

**Abstract:** Advances in remote sensing technologies have enabled effective drought monitoring globally, even in data-limited areas. However, the negative impact of drought on crop yields still necessitates stakeholders to make informed decisions according to its severity. This research proposes an algorithm to combine a drought monitoring model, based on rainfall, land surface temperature (LST), and normalized difference vegetation index/leaf area index (NDVI/LAI) satellite products, with a crop simulation model to assess drought impact on rice yields in Thailand. Typical crop simulation models can provide yield information, but the requirement for a complicated set of inputs prohibits their potential due to insufficient data. This work utilizes a rice crop simulation model called the Simulation Model for Use with Remote Sensing (SIMRIW–RS), whose inputs can mostly be satisfied by such satellite products. Based on experimental data collected during the 2018/19 crop seasons, this approach can successfully provide a drought monitoring function as well as effectively estimate the rice yield with mean absolute percentage error (MAPE) around 5%. In addition, we show that SIMRIW–RS can reasonably predict the rice yield when historical weather data is available. In effect, this research contributes a methodology to assess the drought impact on rice yields on a farm to regional scale, relevant to crop insurance and adaptation schemes to mitigate climate change.

**Keywords:** crop simulation; drought assessment; LAI; LST; NDVI; yield estimation

## 1. Introduction

For the last three decades, the world has experienced increasing occurrences of various disastrous events whose impacts on the economy cost approximately 250–300 billion USD annually [1]. In terms of agriculture-related disasters, the top three causes are droughts, volcanic eruptions, and storms, which are 83%, 30%, and 23%, respectively [2]. Therefore, the impact of drought can be devastating and must be handled with extreme priority. Thailand, as one of the top rice-exporting countries, is very susceptible to drought, as more than 58% of farmers still have no access to water resources (i.e., rainfed only), and about 60% tend small farms (<2 ha) [3]. Significant droughts (declared drought areas of more than 160,000 ha) have recently been recorded in 2012, 2014, 2015, 2018, and 2019 [3].

In response to this urgent problem, the Thai government initiated a pilot rice crop insurance program for farmers in 2010, aiming to cover loss or damage from major natural disasters including drought. The policy is based on weather index crop insurance [4], where damage is assessed by cumulative rainfall (and compared to a threshold value) from a nearby weather station within a 25km range. This scheme seems suitable for rainfed farmlands because there is no interference from other water resources. Weather indices must be readjusted for each area. However, a limited number of operational weather stations can lead to a high variance in damage assessment, especially for drought, whose nature is a complex interaction among domains of parameters such as soil, weather, crop, and field practices. As a result, this type of insurance policy incurs a large basis risk (i.e., difference in the actual losses compared to the losses projected by the policy). This program is not well received by farmers but is still offered in the northeastern region.

In 2011, the Thai government launched another program called "nationwide relief top-up insurance for rice [5]." The government subsidizes most of the insurance premium, where farmers pay much less to receive full coverage. In addition to the weather index, payouts for total losses are verified by local authorities, normally by visual inspections which hardly cover the actual losses. In order to mitigate this problem, a drought map[1] provided by the Geo-Informatics and Space Technology Development Agency (GISTDA) has been used recently. This geographic information system (GIS)-based service is updated weekly and is based on products from the Suomi–NPP satellite. Although these spatial and temporal data can augment assessment, they cannot accurately capture damage to crop yields, and, hence, this leads to a large number of basis risk complaints.

Previous work [6] stated that a conventional policy can become ineffective to underwrite risks and adjust losses at the field level. It shows that a weather index based on precipitation alone has very little correlation with soil moisture. On the contrary, soil moisture and normalized difference vegetation index (NDVI) are good indicators for drought and good proxies for yield. This study mentioned that big data analytics, such as text/visual analytics, predictive modeling, and machine learning are becoming essential tools for agriculture insurance. Big data includes weather and climate data (from satellites, drones and weather stations), soil and geo–spatial data (from mobile apps, location datasets, satellites, drones and ground sensors) and crop yield (from mobile apps, drones and satellites). The mobile app has become an essential tool for localized data collection through crowdsourcing, in conjunction with advanced satellite remote sensing products, which enable field-level insurance policies to be administered.

Although crop insurance can compensate the drought losses for farmers and shift the burdens from the government to insurance companies, it does not optimize the losses, which could have been reduced if farmers had known in advance about risks to their crop yields. For example, farmers could implement crop changes or an early harvest to adapt to some worst-case scenarios. Without adaptation plans, food security can be compromised if extreme droughts occur (which become more likely under current climate change conditions). In order to mitigate this, crop yield estimation and prediction algorithms, in combination with drought monitoring algorithms, seem inevitable for drought assessment and adaptation [7], for which crop production gets impacted and warnings can be provided promptly. Comprehensive works have been performed on drought monitoring based on satellite products [8–10], and, similarly, active works on rice yield estimation based on crop simulation models (CSMs), such as the Decision Support System for Agrotechnology Transfer (DSSAT) [11], the World Food Studies Simulation Model (WOFOST) [12], the Cropping Systems Simulation Model (CropSyst) [13], AquaCrop [14] and the Simulation Model for Use with Remote Sensing (SIMRIW–RS) [15]. Nevertheless, to our best knowledge, there are no such works to demonstrate how they can effectively work together. A major disparity between the two is their coverage. Drought monitoring models are normally regional scale, utilizing various types of satellite remote sensing indices, whereas CSMs require more input parameters and, hence, work only on specific areas where such input data are available. These inputs can be categorized as cultivar specific, soil specific, farm management, and climatic inputs that are obtained by field data collection. Climatic inputs need to be continuously applied until the crop ends (usually by a weather

---

[1] Available at http://droughtv2.gistda.or.th/?q=content/drought–mapping

station nearby) to complete the simulation. The common simulated outputs from CSMs are leaf area index or canopy cover, which are converted to biomass and yield at the final step. The other inputs are initial conditions, a few of which can be readjusted during the crop cycle to recalibrate the simulated results to be close to the observed data from the field. The authors of [14] showed how to use the HJ–1A/B NDVI satellite product to readjust the maximum canopy cover parameter in AquaCrop to improve the accuracy of the simulated rice yield. While satellite NDVI products can potentially enable regional-scale simulation in AquaCrop, this work still depends largely on field data collections and data from weather stations, limiting its adoption to other areas. A research question then becomes: can satellite products be applied to both drought monitoring models and crop simulation models so that they can work cooperatively and effectively at a farm scale, can they be extended to a regional scale, and how can this objective be acheived?

This research answers the question in four aspects. First, we evaluate the effectiveness of existing satellite-based drought monitoring models [16]. Second, we propose to utilize the same satellite products, in place of data from weather stations, as climatic inputs into a crop simulation model called SIMRIW–RS. Third, we introduce a recalibration algorithm for SIMRIW–RS to attain better accuracy using satellite NDVI and leaf area index (LAI) products. Collectively, we show that these satellite data products work very well on the test site, achieving around 5% mean absolute percentage error (MAPE). Fourth, we empirically show how the drought situation can signify the simulated rice yield from SIMRIW-RS. Later, we suggest how this work can be extended to provide nationwide services with existing spatial data from the Thai government agencies and crowdsourcing data from farmers via a mobile app.

## 2. Materials and Methods

This study utilizes data from two main group sources. Figure 1 depicts data collected from the field (shown in blue boxes); as described in Section 2.2, these data are used as inputs for a crop simulation model called SIMRIW–RS to evaluate the bounded performance for the simulated yield versus the actual yield. On the other hand, satellite data products (shown in red boxes), detailed in Section 2.3, are proposed to replace field data for presumably comparable performance, with the potential to extend to the regional scale. Some of these satellite products are also used by the drought monitoring model, thereby allowing integration of drought monitoring and impact assessment on rice yield on the same platform.

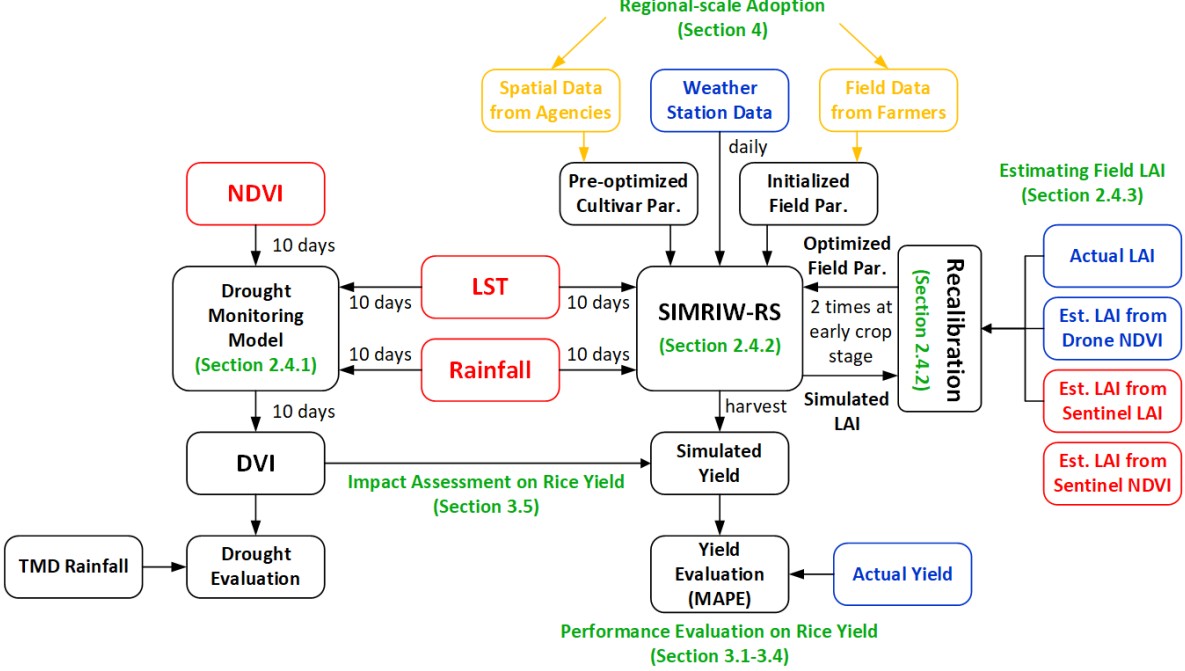

**Figure 1.** Research methodology and data sources used in this study. See text for full details.

This figure also outlines the research methodology and emphasizes how satellite products can effectively be applied to drought monitoring and crop simulation models. Section 2.4.1 evaluates drought monitoring models based on the drought vulnerability index (DVI) and monthly rainfall from the Thai Meteorological Department (TMD) and compare the consistency of the DVI values before and after rainfall. SIMRIW–RS is introduced in Section 2.4.2 along with its required recalibration algorithm to improve the accuracy of the simulated yield. This recalibration requires field data called LAI, which is impractical to collect on the field; hence, Section 2.4.3 elaborates the performance of using satellite products to estimate the actual LAI values. Sections 3.1–3.4 explain simulation configurations with weather station data and satellite products as inputs, and with actual LAI and estimated LAI for recalibrations. Performance for each configuration is measured as % of MAPE against the actual yield. Section 3.5 demonstrates how DVI can explain the variation of the simulated yield from SIMRIW–RS, signifying the working scheme for both models. Section 4 discusses how SIMRIW–RS can potentially be adopted at the regional scale.

## 2.1. Test Site

The selected test site is a 6.88-ha paddy field located in northeast Bangkok (13.93628, 100.87029), as shown in Figure 2. The soil is an acidic clay type with an average annual precipitation of 1721 mm. Irrigation networks allow farmers to cultivate rice twice a year, but water shortage is likely during drought events. Monocrops with sowing practices and chemical fertilizer usage are common in this area. Data were collected during April 26–August 17, 2018 based on the RD57 variety (114-day crop cycle). Fifty sample plots of 1x1m$^2$ were assigned on the periphery to measure LAI and yields and mark the locations for comparison with the remote sensing data products. The data collection equipment were a GPS Garmin Colorado 3000, a LAI–2200 Plant Canopy Analyzer, a DJI Phantom 3 advanced with a Parrot Sequoia multispectral camera installed, and other typical tools for crop cutting and grain moisture analysis, as depicted in Figures 3 and 4.

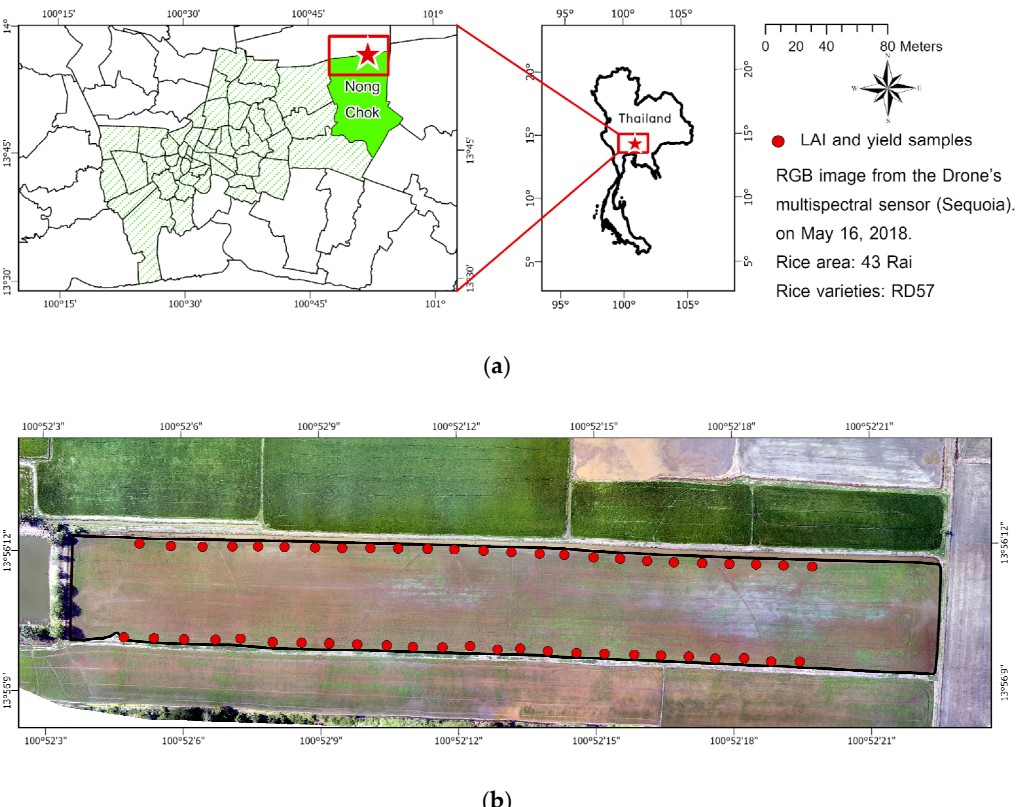

(**a**)

(**b**)

**Figure 2.** (**a**) The test site (indicated by a red star) located in northeast Bangkok. (**b**) True-color image of the site, taken by a DJI Phantom 3 advanced equipped with a Parrot Sequoia multispectral camera at 10 cm ground spatial distance (GSD). Red dots on the periphery represent 50 sample plots.

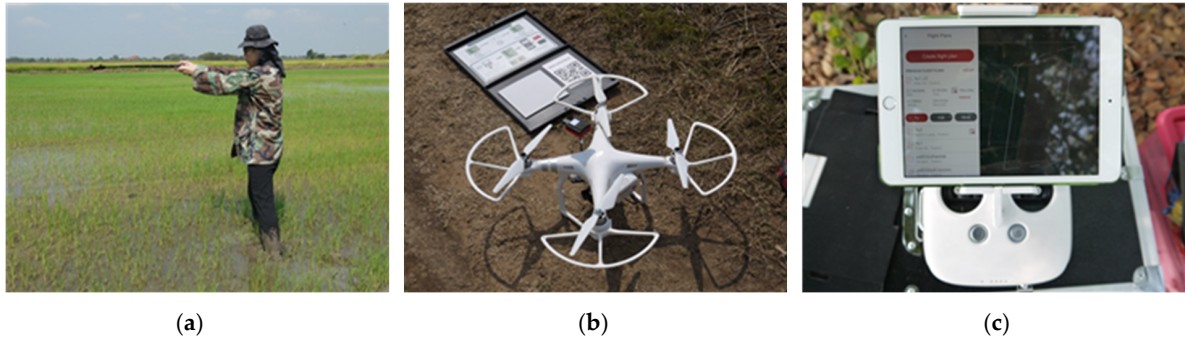

| (**a**) | (**b**) | (**c**) |

**Figure 3.** (**a**) Leaf area index—LAI measurement with a LAI–2200 Plant Canopy Analyzer. (**b**) and (**c**) Data collection by a DJI Phantom 3 with a multispectral camera at the test site.

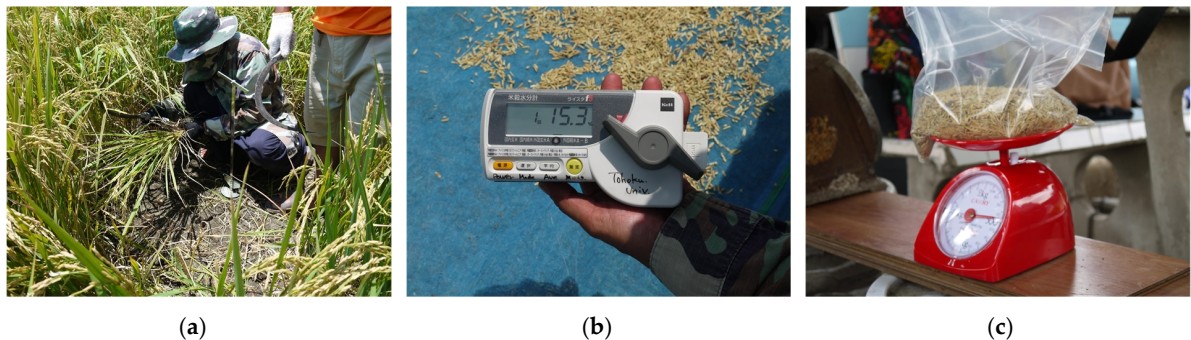

| (**a**) | (**b**) | (**c**) |

**Figure 4.** Yield data collection activities at harvest: (**a**) crop cutting in a 1x1m$^2$ plot, (**b**) measuring grain moisture content, and (**c**) weighting grains at 0% moisture content.

### 2.2. Field Data Collection

To ensure data integrity, field data collections were performed eight times during the entire crop season, as detailed in Table 1. LAI [17] is an index representing plant canopies. It is defined as the one-sided green leaf area per unit ground surface area and is normally used to indicate crop health/growth conditions. Time-series LAI data are very important to understand how crops respond to changes in their environment.

Remote sensing data from drones are extremely important to this study. As we aim to develop a methodology based on satellite data, drone data fill the gap between point-based ground truth data and lower-spatial-resolution (≥10 m) pixel-based data from satellites. Nowadays, drones can be equipped with sensors whose characteristics resemble those installed aboard remote sensing satellites. Thus, variabilities from different spatial and temporal localities of both field and satellite data can be studied and deduced. Figure 5 shows the spectral responses from a Parrot Sequoia multispectral camera and the corresponding Sentinel–2A bands that were used in this work.

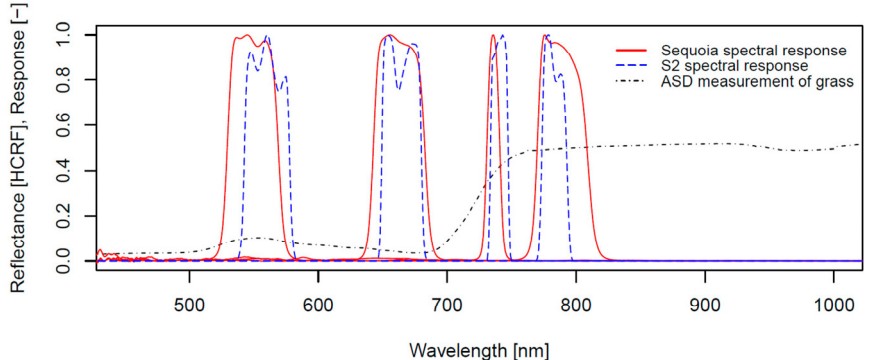

**Figure 5.** Relative spectral responses of a Parrot Sequoia camera (Green, Red, Red Edge and NIR), as well as the most closely corresponding Sentinel–2A bands (bands 3, 4, 6 and 7) (excerpted from [18]).

**Table 1.** Field data collections in 2018.

| No. | Date | Rice Age (weeks) | Average leaf area index—LAI | Average Yield (ton·ha⁻¹) |
|-----|------|------------------|-----------------------------|--------------------------|
| 1 | May 16, 2018 | 3 | 0.83 | – |
| 2 | May 20, 2018 | 5 | 2.79 | – |
| 3 | June 7, 2018 | 7 | 3.94 | – |
| 4 | June 20, 2018 | 8 | 4.54 | – |
| 5 | June 29, 2018 | 10 | 4.67 | – |
| 6 | July 9, 2018 | 11 | 4.70 | – |
| 7 | August 1, 2018 | 14 | 4.36 | – |
| 8 | August 14, 2018 | 16 | – | 4.35 |

Weather data were collected from an automatic weather station located 2.55 km from the test site (13.918123, 100.855946). The station automatically transmits data at 10-minute intervals to a server at the Chulabhorn Satellite Receiving Station (CSRS) at Kasetsart University. These are processed as daily data, including accumulated rainfall (mm), average solar radiation (W·m⁻²), average temperature (°C), and day length (hours). These field data are utilized only for model development purposes, as they will be substituted by the equivalent satellite products when they are operational.

Yield data were collected at harvest time (when grain moisture contents are about 15–20%) on all 50 plots. Grain and straw were separated and dried under sunlight until the moisture contents were reduced to 15%. These specimens were taken to a lab to be machine dried at 70 °C until the weights were stable, which is represented by the weights at 0% moisture content. The average weights at 0% moisture content for grain and straw were 4.35 ton·ha⁻¹ and 6.78 ton·ha⁻¹, respectively.

*2.3. Satellite Products*

Satellite products used in this work consisted of two groups: weather-related and crop-related products. The first group comprises rainfall and land surface temperature (LST), whereas the other group consists of NDVI and LAI. To address the availability and quality issues in satellite data, weather-related products can be updated every ten days, called "dekadal" products. On the other hand, crop-related products are updated as necessary, subject to their functions, such as an input to a drought monitoring model or as a recalibration parameter for a crop simulation model. These satellite products, except for Sentinel–2, are received directly at the CSRS in real time.

2.3.1. Rainfall from the FY–2E Satellite

FY–2E is one of the geostationary meteorological satellite constellations operated by China Meteorological Agency (CMA). Our previous work [19] developed a model called Infrared Threshold Rainfall with Probability Matching (ITRPM) to estimate hourly rainfall (known as FY–2 rain) using data from the IR–1 channel (10.3–11.3μm) with a 5-km spatial resolution, shown in Figure 6a. The model was optimized for the Thailand region and has shown good root-mean-square error (RMSE) for medium- to long-term accumulated rainfall. Figure 6b shows the final product, called dekadal rainfall, which represents total precipitation during ten consecutive days.

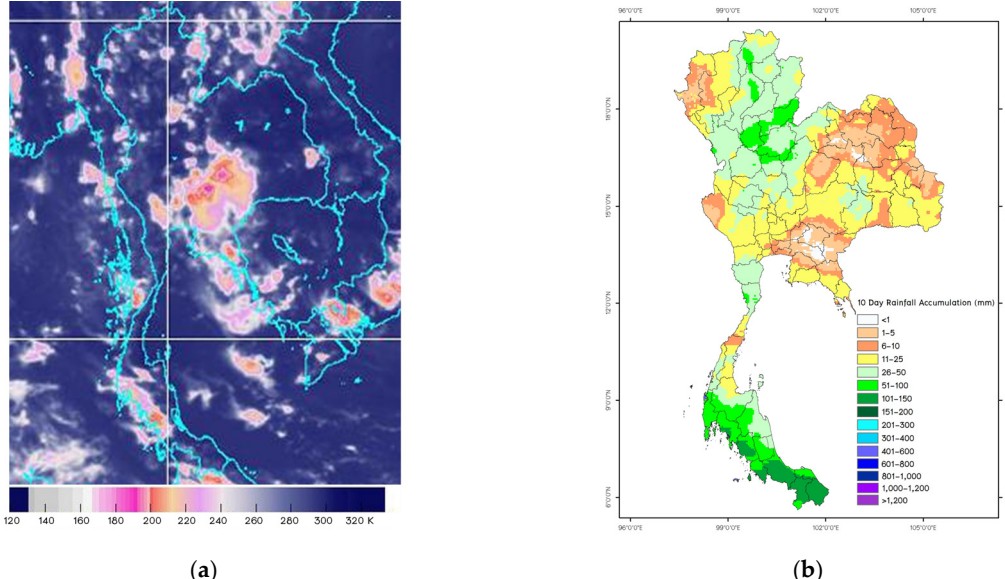

| (**a**) | (**b**) |

**Figure 6.** Examples of (**a**) IR–1 image and (**b**) dekadal (10-day) rainfall product.

### 2.3.2. LST from TERRA/AQUA Satellites

Land surface temperature (LST) exhibits a good relationship with dynamic soil moisture in water-limited conditions [20] and also with aboveground temperature where the leaf temperature rises due to closure of plants' stomata to preserve root-zone water loss through transpiration and soil temperature increases due to the ceasing of evaporation. This work selects a product called Land Surface Temperature/Emissivity Daily L3 Global (coded MOD11A1) based on the Moderate Resolution Imaging Spectroradiometer (MODIS) aboard the TERRA/AQUA satellites. The product has 1-km spatial resolution data and is available twice a day (day and night), as depicted in Figure 7. These day and night data are averaged over a 10-day period and are converted to the maximum and minimum air temperatures, respectively.

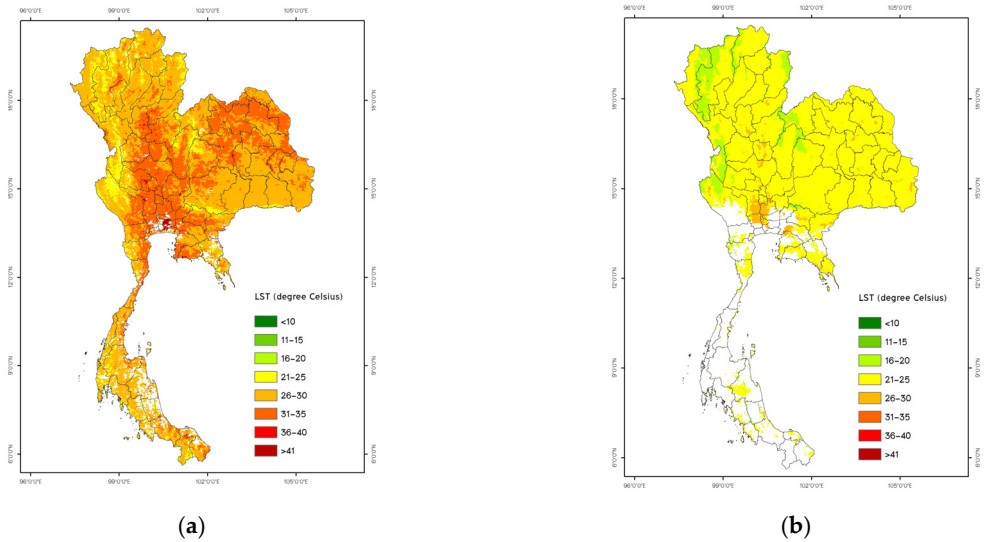

| (**a**) | (**b**) |

**Figure 7.** Examples of land surface temperature (**a**) LST—day product and (**b**) LST—night product on October 29, 2019.

### 2.3.3. NDVI from TERRA/AQUA satellites

The normalized difference vegetation index (NDVI) is commonly used as an interpretation of photosynthesis activity in plants. It can indicate crop stress that relates to depleted available

moisture in the root zone [9], implying a drought situation. This work selects a product derived from MODIS's Surface Reflectance Daily L2G Global product (coded MOD09GA) with a 500-m resolution, using Band 1 (visible red or RED) and Band 2 (near-infrared or NIR) according to the following equation:

$$NDVI = (NIR - RED) / (NIR + RED) \qquad (1)$$

The result is then cloud-masked and averaged over a ten-day interval. To standardize these satellite products derived in Sections 2.3.1–2.3.3 for ease of implementation, they are resampled to a 1-km resolution, time-synchronized, and put into archive to be used as inputs for both models. Related work [21] shows reasonably good correlations between the satellite rainfall and LST products with rainfall and average air temperature from TMD weather stations nationwide.

### 2.3.4. NDVI/LAI from the Sentinel–2 Satellite

NDVI and LAI products are derived from the Multi-Spectral Instrument (MSI), covering visible to shortwave infrared bands with 10-m, 20-m and 60-m spatial resolutions on the Sentinel–2A/B satellites. Its level-1C data (called S2MSI1C)[2] are top-of-atmosphere (TOA) reflectance and need to be processed with the Sentinel Application Platform (SNAP)[3] and Sen2Cor[4] plugin to generate level 2A data. The level 2A data consist of bottom-of-atmosphere (BOA) reflectance and scene classification (SCL). Finally, the NDVI and LAI products are derived from BOA reflectance using NDVI and biophysical processors on SNAP, and then cloud–masked by SCL, as depicted in Figure 8. Both NDVI and LAI products have a 10-m resolution and can be updated every 5 days.

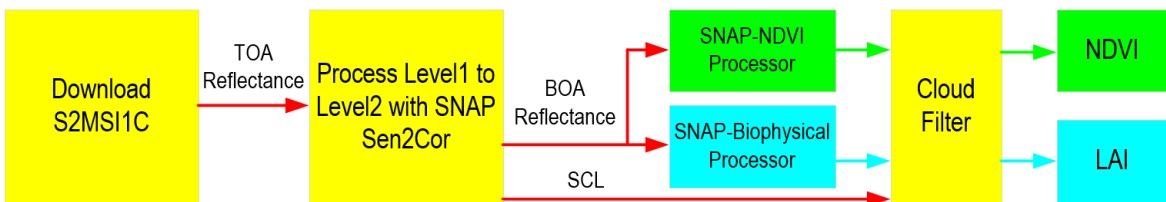

**Figure 8.** Data processing for Sentinel–2's NDVI/LAI product generation.

### *2.4. Algorithm Development*

We first demonstrate that previous work on drought monitoring models [19] based on weather- and crop-related satellite products can successfully capture the essence of both meteorological and agricultural droughts, collectively known as a drought hazard index (DHI). These spatial data, when overlaid with agricultural land use from the Land Development Department (LDD) and irrigation data from the Royal Irrigation Department (RID), become a drought vulnerability map[5] to be used for efficient drought management with regard to water resources and crop production. Consequently, we show how those satellite products can be linked to a crop simulation model to estimate or even reasonably predict crop yield under drought conditions. These comprehensive features have been considered one of the key challenges to the future agricultural drought monitoring system [7].

### 2.4.1. Drought Monitoring Model Evaluation

In ref. [16], three satellite products were used for drought monitoring. Rainfall estimates from the FY–2E satellite represent meteorological drought from a precipitation deficit. The difference between day and night LST indirectly represents the surface soil moisture indicator, which is a good proxy of if a planted crop can reach its full potential. The NDVI represents the photosynthesis

---

[2] Available at https://scihub.copernicus.eu/dhus/#/home
[3] Available at https://step.esa.int/main/toolboxes/snap/
[4] Available at https://step.esa.int/main/third-party-plugins-2/sen2cor/
[5] Available at http://csrs.ku.ac.th/wegis/product/adap–t

activity of plants, an indicator for crop health and conditions that represents agricultural drought. Both are TERRA/AQUA products. Because drought is a slow-onset phenomenon, this model updates results every ten days using dekadal values, which are accumulated rainfall, average difference LST and average NDVI. A linear function of these values, optimized for the Thailand region, generates what is called the drought hazard value:

$$DH = 11.643 + 0.356 \cdot R - 0.297 \cdot \Delta T + 0.868 \cdot NDVI_{AVG} \tag{2}$$

where DH is the drought hazard value, R is the accumulated rainfall, $\Delta T$ is the average difference LST, and $NDVI_{AVG}$ is the average NDVI, all for each 1-km$^2$ pixel resolution. All DH values from 2016 were sorted and arranged to form a cumulative distribution function. Threshold values were defined to partition the DH values into seven percentile groups; for example, if DH values are less than the fifth percentile, this can be considered an extreme drought condition. Meanwhile, a DH value greater than the 95th percentile is classified as an extreme wet condition. DHI, ranging from –3 (extreme drought) to +3 (extreme wet), was defined based on those percentile groups. However, DHI alone does not explain the drought situation as it does not account for agricultural activities and countermeasures against drought in the area concerned. A more informative index, called the drought vulnerability index (DVI), is defined as a composite of DHI, agricultural areas, and irrigated areas, to show which agricultural areas are really affected by drought. We first define the sensitivity index (SI) as the ratio of agricultural areas to the total areas. Irrigation networks can ameliorate drought situations and create adaptive capacity (AC) to drought. Both SI and AC are based on provincial data. Their relationships can be described as follows:

$$SI = \text{agricultural areas / total areas} \tag{3}$$

$$AC = 1 - (\text{irrigation areas / total areas}) \tag{4}$$

$$DVI = DHI \times SI \times AC \tag{5}$$

Figure 9 shows the DVI for Thailand in March 2020 and May 2020 versus monthly rainfall in April 2020 from TMD. This figure clearly shows the effectiveness of the drought monitoring model, where most parts of Thailand in March 2020 were experiencing extreme drought (shown in red) except the northern, western, and lower central regions. Precipitation in April 2020 ameliorated the extreme drought conditions in those areas a month later when farmers could start their crops, especially in the northeastern, eastern, and southern parts of Thailand. Those areas are mostly rainfed paddy fields.

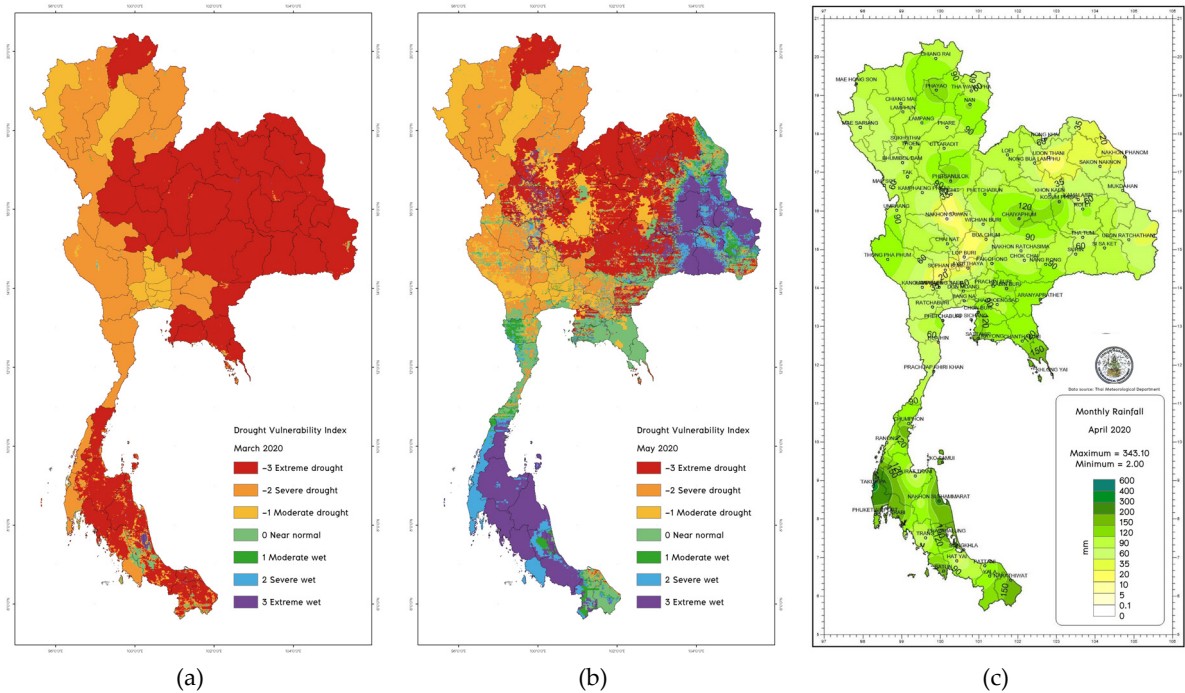

|  | (a) | (b) | (c) |

**Figure 9.** The drought vulnerability index for Thailand in (**a**) March 2020 and (**b**) May 2020, and (**c**) monthly rainfall in April 2020. Precipitation areas are relatively consistent with improved drought situations.

### 2.4.2. SIMRIW–RS and Recalibration Method

SIMRIW–RS [15] is a crop simulation model for rice that requires only three sets of inputs. First, the daily weather dataset, which includes rainfall, average temperature, day length, and average solar radiation. Second, the pre-optimized cultivar parameters, which are localized data embedded in the model and can be estimated based on historical yields and the corresponding NDVI or LAI time-series data. However, these cultivar parameters are not currently user-adjustable. Third, the initial field parameters, which need to be optimized by recalibration of the simulated LAI (an output from the model) with the actual LAI value from the field at an early crop stage. Since LAI measurements are impractical for model operations, satellite products are highly recommended for LAI estimations: [22] shows that two recalibrations, at early crop stages, of the simulated LAI with synthetic aperture radar (SAR)-estimated LAI can achieve good results. Similarly, [23] shows that two recalibrations within 50 days after sowing, using drone NDVI or two-band enhanced vegetation index (EVI2) products to estimate LAI, are also possible. The flexible recalibration dates allow satellite products to be used for LAI estimation, particularly the NDVI/LAI products from Sentinel–2 satellites with a 10-m resolution and 5-day revisit time.

An algorithm for SIMRIW–RS recalibrations, using datasets from 2018, is illustrated in Figure 10. The daily weather data were intentionally replaced by dekadal satellite products. Rainfall from the FY–2E satellite and LST from the TERRA/AQUA satellites were averaged daily and used as rainfall and temperature inputs. The day length was calculated with software developed with Python and the ephem package[6]. Lastly, the solar radiation was obtained from a nearby weather station. The cultivar parameters were optimized for the test site in 2018.

With a daily weather dataset input being ready, SIMRIW–RS can start simulating the rice crop with its default parameters. It generates the daily simulated LAI output, simulated dry matter and simulated yield at the crop end. The simulated LAI output corresponds to the nitrogen uptake ability of the field, which is one of the most relevant field parameters needed for accurate crop simulation. SIMRIW–RS lets the user optimize this parameter (ranging from 0.040 to 0.011 with 0.0001 steps) through recalibration of its simulated LAI with an estimated LAI from the Sentinel–2

---

[6] Available at https://pypi.org/project/ephem/

LAI product using a numerical method, which iterates until the RMSE of the simulated LAI is minimized. Once the parameter is ready, the model can simulate till crop harvest, and its performance is measured as the percentage MAPE of the simulated yield against the actual yield.

Since there are many estimated LAI values for the test site (each value corresponds to a 5x5m² area), we considered how SIMRIW–RW should handle these data. The first approach is to let SIMRIW–RS handle them individually and average the results to obtain the simulated yield. The second approach assumes a homogenous LAI value by averaging all the data and then lets SIMRIW–RS handle this data to obtain the simulated yield. The MAPE of the simulated yield will determine which approach is more efficient, as the first approach requires more computation.

Of the three sets of input parameters, the cultivar parameters (pre-optimized for 2018 crop data) were embedded in the model and are not adjustable. Therefore, we used SIMRIW–RS to perform the simulation on the test site with 2019 crop data (all other inputs are from satellite products, except the actual yield) to observe how robust these cultivar parameters are when other parameters changes. This also validates our proposed algorithm.

Finally, we evaluated the ability of SIMRIW–RS to reasonably forecast the yield by preparing the future weather dataset from historical satellite products. This weather data will be used, after the recalibration process is performed, by SIMRIW–RS to complete the simulation for two months earlier. The output simulated yield is, therefore, known to the user beforehand, enabling farmers to make informed decisions on adaptation to climate change.

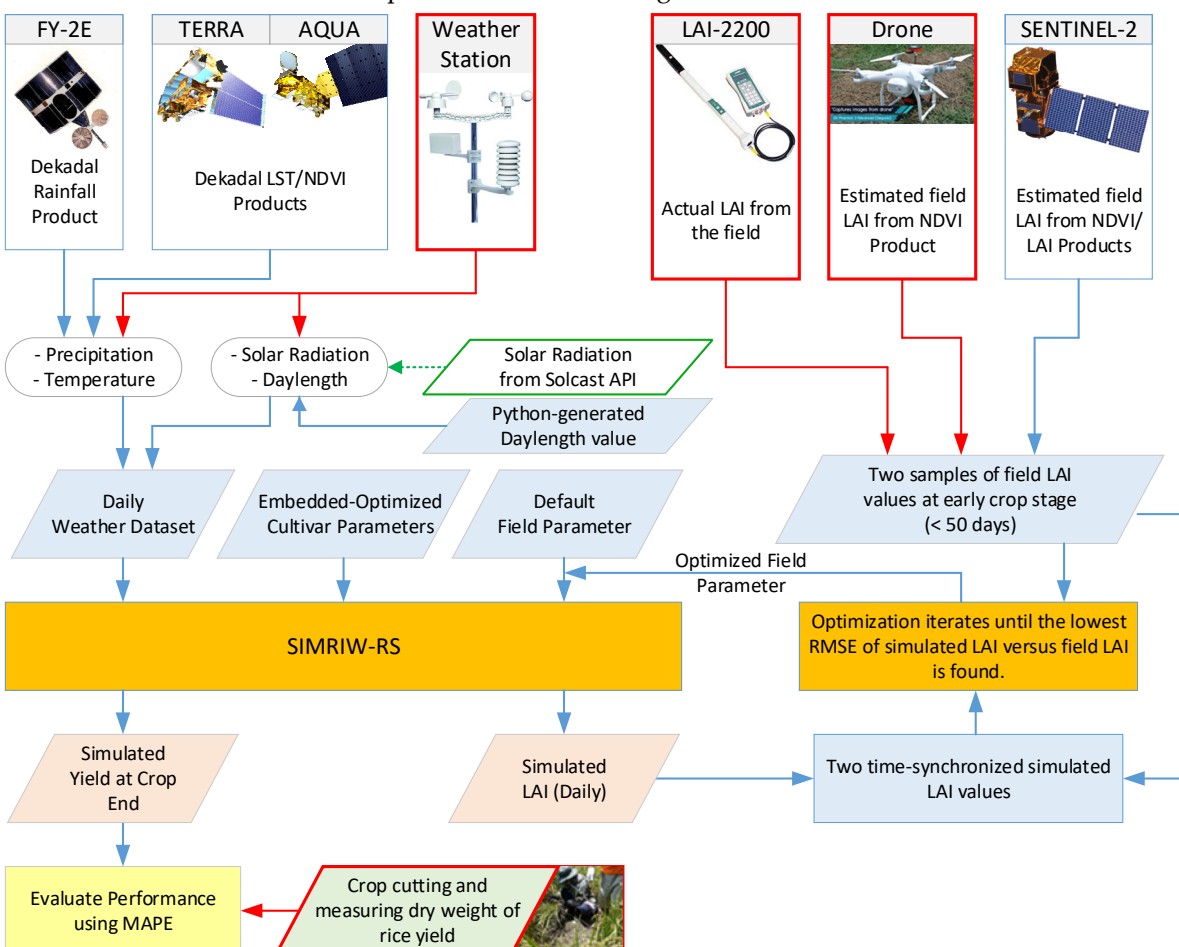

**Figure 10.** Proposed algorithm for the Simulation Model for Use with Remote Sensing—SIMRIW–RS recalibration. Red boxes represent data sources for recalibration and validation processes only, whereas green boxes represent future work. Only satellite products are required when operating SIMRIW–RS.

2.4.3. Estimating Actual LAI with Remote Sensing Data

This study proposes to use Sentinel−2 NDVI/LAI products to estimate the actual LAI values required by the SIMRIW−RS recalibration process. Their performances were evaluated against that of the drone NDVI product. The LAI values and multispectral data from the drone were collected at the test site. Each LAI value is indeed an average of five LAI measurements in a 1x1m$^2$ plot. The drone data from a Sequoia multispectral camera were processed with Pix4D software to generate NDVI products with a 10-cm ground sample distance (GSD) and were plotted against the actual LAI on the same locations as shown in Figure 11. Since the drone pixel is 100 times smaller than a plot, its NDVI products are averaged to match the actual LAI value. Only the first three data collections (corresponding to early crop stages) were considered for a total of 150 data pairs. This should give a performance bound on estimating actual LAI with Sentinel–2 products. It shows an exponential relationship with $R^2 = 0.8336$, meaning that the drone NDVI can reasonably estimate the LAI from the field. The equation is:

$$Y = 0.1524e^{3.4927 \cdot X} \tag{6}$$

where Y is the estimated LAI and X is the drone NDVI product.

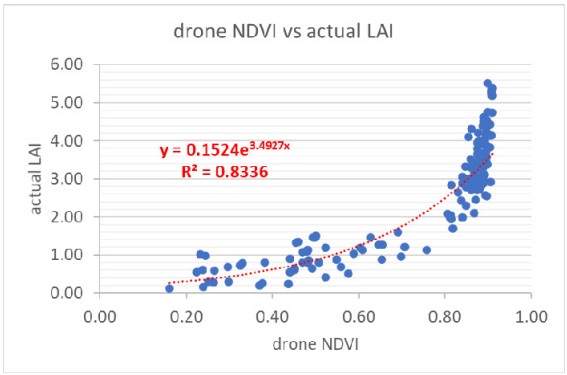

**Figure 11.** Exponential relationship (with $R^2 = 0.8336$) between drone NDVI and actual LAI values at early crop stages.

NDVI products from TERRA/AQUA and Sentinel–2 might be used. However, at a 10-m resolution and 5-day revisit time, the Sentinel–2 NDVI product is superior in a farm-level simulation without compromising the recalibration performance. The Sentinel–2 LAI product is also investigated. The Sentinel–2 data were acquired on May 22, May 26 and June 6, 2018, respectively. We first compared the relationships between the Sentinel–2 NDVI and LAI products with the actual LAI in Figure 12. Only the pixels that collocated with the sample plots were considered, totaling 68 data pairs for each product. The Sentinel–2 LAI product showed a promising result for LAI estimation with $R^2 = 0.8364$. In contrast, the Sentinel–2 NDVI product performed much worse with $R^2 = 0.4622$.



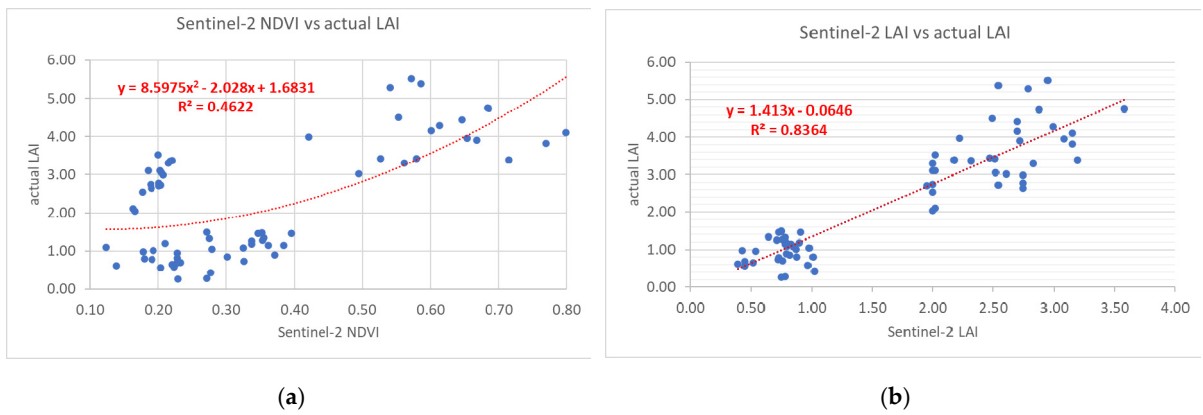

(**a**)                                                                    (**b**)

**Figure 12.** Performance of LAI estimation using (**a**) Sentinel–2's NDVI product and (**b**) Sentinel–2's LAI product.

Direct estimation of LAI from Sentinel–2 NDVI constitutes pixel size and index disparities. Thus, we split this process into two steps. First, a relationship between the Sentinel–2 NDVI and drone NDVI was found; see Figure 13a. This gave us insight into the effect of pixel size disparity. Second, the estimated drone NDVI was used to estimate the LAI using the relationship obtained from Figure 11. The derived product is called "Sentinel–2 NDVI–estimated LAI". Figure 13b shows that the overall performance of $R^2$ = 0.4185 is comparable to the direct estimation, $R^2$ = 0.4622, from the Sentinel–2 NDVI. We noticed that the Sentinel–2 NDVI products were very dispersed at values < 0.5, when the rice age was less than 5 weeks and its surface was widely covered with water. In addition, since our plots were on the periphery of the test site, the collocated Sentinel–2 pixels may include data from nearby paddy fields. Both cases may account for the dispersion of Sentinel–2 NDVI values. We also plan to investigate other satellite NDVI products for more conclusive results. To ensure robust operation, it is quite common to prepare for alternative satellite NDVI products when the Sentinel–2 LAI product is not available. It is clear that the Sentinel–2 LAI product can reasonably estimate the actual LAI for the next step. The equation for the estimated LAI is

$$Y = 1.1413 \cdot X - 0.0646 \tag{7}$$

where Y is the estimated LAI and X is the Sentinel–2 LAI product.

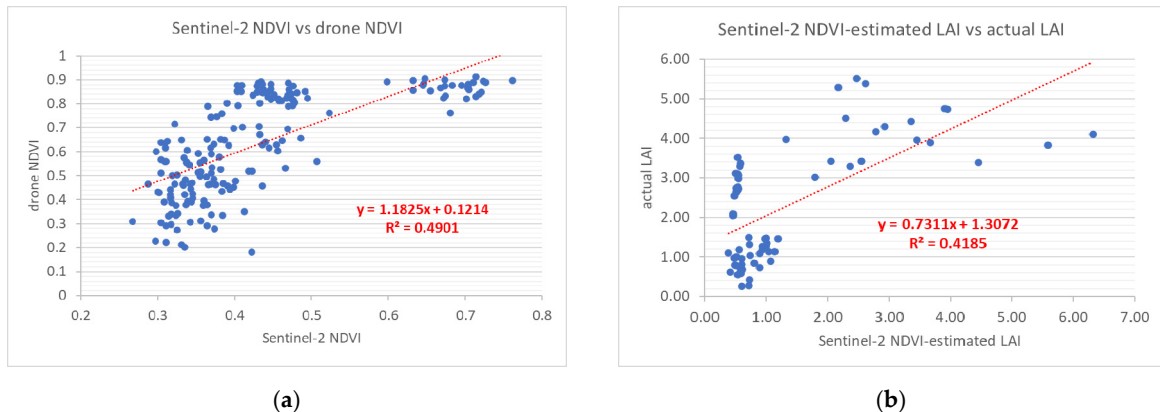

(**a**)                                                                    (**b**)

**Figure 13.** A two-step conversion relationship from Sentinel–2 NDVI to (**a**) drone NDVI and (**b**) actual LAI, with a total $R^2$ = 0.4185. Sentinel–2 NDVI were highly dispersed at values < 0.5, yielding poor conversion in the first step and affecting the overall performance.

## 3. Simulation Results

### 3.1. Performance Based on Weather Station Data

These experiments utilized data from a weather station to provide best-case scenarios. They are summarized in Table 2. The LAI values were obtained from field measurements (LAI2200), and the NDVI product from the Parrot Sequoia Multispectral Camera (Drone) and from the Sentinel–2 LAI product (Sentinel-2), called E1, E2 and E3, respectively. There were two simulation modes, homogenous and non–homogenous. The homogeneous mode assumed a single LAI value averaging over all the LAI data and SIMRIW–RS was executed only once for the test site. This should work well for small farms with monocrop practices. The non-homogeneous mode executed on all LAI data. Performance was measured as the MAPE of the simulated yield versus the actual yield. All homogeneous simulations performed better than their counterparts, which might be a result of the law of averages, with the best MAPE at 3.61% from LAI–2200 (E1). Surprisingly, the MAPE from Sentinel–2 (E3) was the second best at 6.45%, slightly better than the MAPE at 6.75% from drone (E2). This result confirms our previous suggestion that Sentinel–2 LAI is better than drone NDVI for estimating the LAI.

**Table 2.** Performance of SIMRIW–RS based on weather station data.

| Experiment No. | LAI data source | Homogenous LAI | MAPE (%) |
|---|---|---|---|
| E1 | LAI2200 | No | 10.95 |
| | LAI2200 | Yes | 3.61 |
| E2 | Drone | No | 12.73 |
| | Drone | Yes | 6.75 |
| E3 | Sentinel–2 | No | 10.10 |
| | Sentinel–2 | Yes | 6.45 |

### 3.2. Performance Based on Satellite Products

These experiments substituted weather station data with dekadal rainfall and LST products, as shown in Table 3. Note that the solar radiation was still retrieved by a weather station (a substitution with satellite products is underway). It is quite interesting that all results were in favor of satellite products which updated data every ten days. This may suggest that SIMRIW–RS is not very sensitive to weather data. Once again, homogeneous LAI from the field (E4) performed the best with a MAPE of 2.03%, and that from the Sentinel–2 (E6) followed with a MAPE of 5.12%

**Table 3.** Performance of SIMRIW–RS based on dekadal satellite weather-related products.

| Experiment No. | LAI data source | Homogenous LAI | MAPE (%) |
|---|---|---|---|
| E4 | LAI2200 | No | 10.59 |
| | LAI2200 | Yes | 2.03 |
| E5 | Drone | No | 12.34 |
| | Drone | Yes | 5.38 |
| E6 | Sentinel–2 | No | 9.79 |
| | Sentinel–2 | Yes | 5.12 |

### 3.3. Performance Validation for 2019 Crop Data

We validated our algorithm with crop data in 2019. This experiment also proved the robustness of the pre-optimized cultivar parameters embedded in SIMRIW–RS. The rice variety was changed to Phitsanulok 2 (planted between May 4 and August 20, 2019) whose actual yield was 4.90 ton·ha$^{-1}$ (obtained via farmer interview). Weather station data were included in the experiment to evaluate if the model could still perform well with satellite products (as it did with 2018 crop data). The actual LAI were estimated with the Sentinel–2 LAI product to demonstrate the intended satellite-based operation. Table 4 shows that the results were consistent with previous experiments in favor of satellite products and a homogenous LAI value (E8), with the best MAPE at 3.54%.

**Table 4.** Performance validation of SIMRIW–RS for 2019 crop data.

| Experiment No. | Weather Data | Homogenous LAI | MAPE (%) |
|---|---|---|---|
| E7 | Station | No | 7.25 |
| | Station | Yes | 4.05 |
| E8 | Satellite | No | 4.83 |
| | Satellite | Yes | 3.54 |

### 3.4. Yield Prediction Performance for 2018/19 Crop Data

Since the validity of SIMRIW–RS was confirmed, we conducted experiments to evaluate if SIMRIW–RS could reasonably predict the yield in advance, after it was properly recalibrated. The experiments in Section 3.2 and 3.3 (E4 – E6 and E8) were repeated with the current (2018/2019) and 5-year historical (2013–2017) weather datasets from satellite products, as shown in Table 5. Current weather datasets were used until the recalibration of SIMRIW–RS was completed. The 5-year historical satellite-derived data were averaged and applied afterwards. Only experiments with homogenous LAI were conducted, as they showed superior performance. The results showed that SIMRIW–RS calibrated with the Sentinel–2 LAI product worked well with the historical weather dataset from the satellite, yielding an MAPE closed to 5% for both 2018 and 2019. In other words, SIMRIW–RS, once properly calibrated, can not only estimate rice yield at the harvest time but can also forecast the yield about two months before harvest with the historical weather dataset.

**Table 5.** Performance of SIMRIW–RS based on a 5-year average historical satellite-derived weather dataset.

| Experiment No. | Year | LAI data source | MAPE (%) |
|---|---|---|---|
| E7 | 2018 | LAI2200 | 1.99 |
| E8 | 2018 | Drone | 5.34 |
| E9 | 2018 | Sentinel–2 | 5.08 |
| E10 | 2019 | Sentinel–2 | 5.02 |

### 3.5. Combining Drought Monitoring with SIMRIW–RS Crop Yield Estimation

Since the same dataset of satellite products for drought monitoring systems is used as an input weather dataset for SIMRIW–RS, their outputs should show some consistency; i.e., drought may result in a rice yield deficit, whereas more water at certain rice stages may boost the rice yields. Figure 14 depicts the time-series DVI values at the test site during the 2018 and 2019 crop seasons. Since DVI values indicate more water in 2019 during the seedling and booting stages, SIMRIW–RS shows a similar trend with a higher simulated yield at 4.73 ton·ha$^{-1}$ as compared with 4.13 ton·ha$^{-1}$ from the previous year. We can combine both functions to monitor areas with high values of drought vulnerability index and then automatically assess drought impact on their simulated yields. At present, the proposed satellite-based configuration can achieve MAPEs as good as 3.54% and 5.02% for yield estimation and yield prediction, respectively.

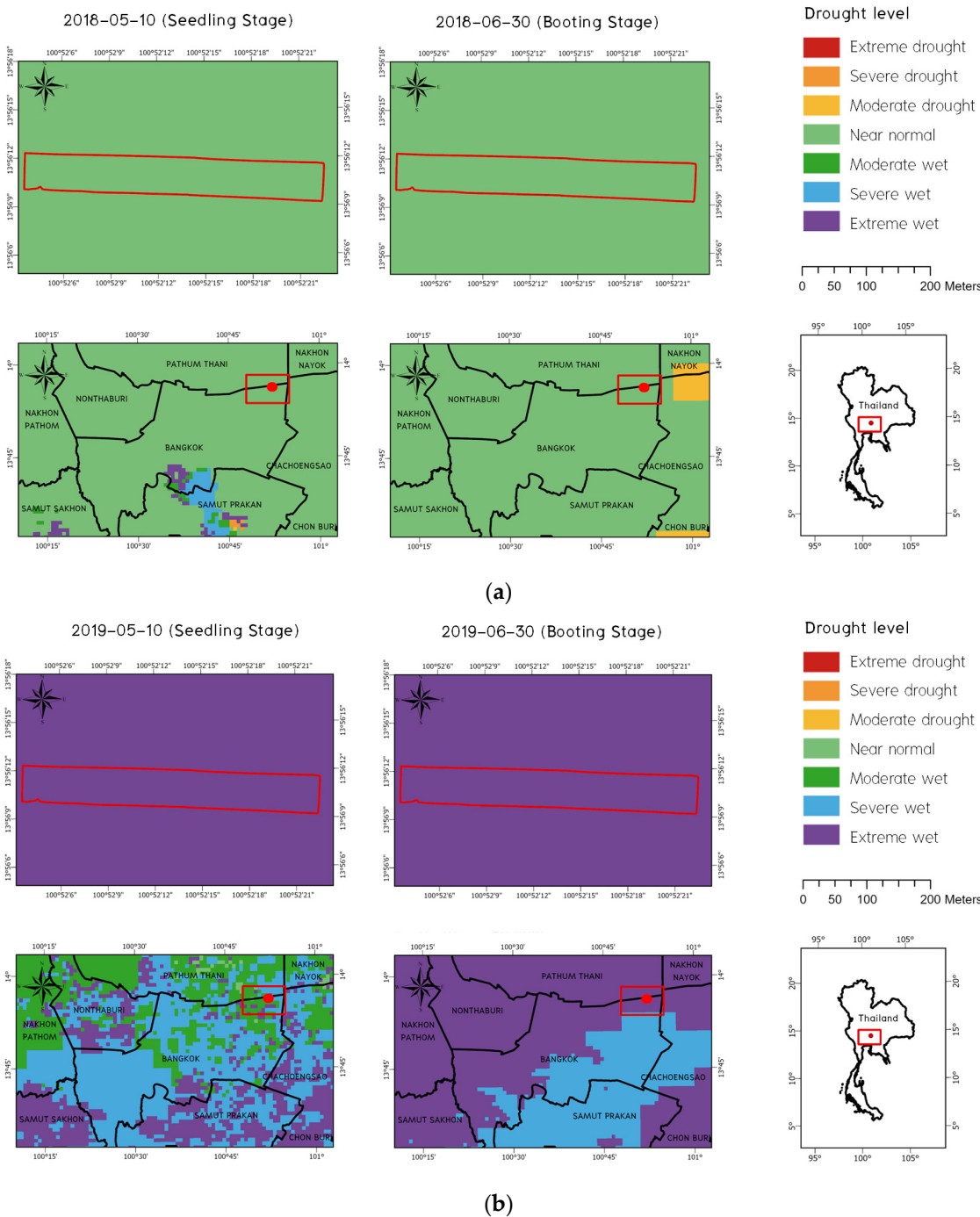

**Figure 14.** Time series (seedling and booting stages) of drought vulnerability index of the test site in (**a**) the 2018 crop season (near normal) and (**b**) the 2019 crop season (extreme wet). Higher wetness in the 2019 crop season may be the reason for the 14.5% higher simulated yield (4.73 ton·ha$^{-1}$) compared with the result from the previous year (4.13 ton·ha$^{-1}$). For each figure, the top figures depict the site periphery whereas the bottom figures depict its location on the map.

## 4. Discussion

Simulation results show that SIMRIW–RS works well with satellite products. Sentinel–2 LAI can be used effectively to estimate actual LAI on the field and is important for recalibrating the field parameter in SIMRIW–RS. The recalibration process is key step in SIMRIW–RS simulation, which must be performed twice within 50 days after planting. Although there are many values of estimated LAI, we confirmed that only the average LAI value of the field can be used to optimize computation while retaining satisfactory results.

Weather datasets from satellite products, e.g., rainfall and LST, though updated every ten days, show results that are somewhat better than those of the weather station (about 1.5% less MAPE). Hence, SIMRIW–RS is not very sensitive to weather data. This characteristic is proven to be good for SIMRIW–RS to work with historical weather datasets for future yield forecasting. The cultivar parameters that were pre-optimized and embedded in SIMRIW–RS revealed themselves to be robust enough to be reused in 2019. Notably, SIMRIW–RS can capture farm-level characteristics through spatial data points in the field. Since these spatial data points can be provided by satellite products, SIMRIW–RS can also operate at a regional scale to serve thousands of users simultaneously. The abovementioned characteristics seem to address our research hypothesis that SIMRIW–RS is a potential crop simulation tool for drought impact assessment on rice yield at a regional scale.

There are three challenges to be solved before SIMRIW–RS can operate nationwide. First, solar radiation input is currently retrieved from a weather station. One solution would be the solar radiation product from a geostationary satellite provided by Solcast[7]. This product is available in real time or as a 7-day forecast with 1-km spatial resolution and is updated every ten minutes. Second, cultivar parameters are currently pre-optimized by the SIMRIW–RS developer using various field and satellite products and embedded into the model. This simplifies employing SIMRIW−RS on the optimized field, with few inputs needed, whereas adaptation to other areas can be laborious. In Thailand, these field and cultivar parameters are available through respective government agency websites. For example, land uses and soil types are available from the Land Development Department, rice varieties from the Rice Department, farmer registration and crop activities from the Department of Agricultural Extension, crop yield and agricultural statistics from the Office of Agricultural Economics, agricultural pests and diseases from the Department of Agriculture, etc. The collection of these data in conjunction with machine learning tools may be a viable solution to automatically localize optimization of the cultivar parameters for the whole country.

The last challenge is to develop a cloud-based smart agriculture platform that incorporates all drought monitoring and SIMRIW–RS functions, including access to the relevant data (IoT/remote sensing/web-based/crowdsourcing), recalibration of the simulated LAI, optimization of the cultivar parameter, and simulation of the entire crop cycle. More importantly, a mobile app is required to provide an intuitive user experience/user interface (UX/UI). This app collects inputs from farmers (crowdsourcing) and displays simulation outputs as well as precautions for some scenarios. To illustrate the concept, Figure 15 shows a related project called "RiceSAP"—a mobile app for rice crop monitoring and management for farmers [21]. The app acts as a user interface to the cloud platform. It currently employs the drought monitoring system and a crop simulation model from the Food and Agriculture Organization of the United Nations (FAO) called AquaCrop[8]. SIMRIW–RS can also be integrated into this platform as another efficient crop simulation model to share the same framework, inputs, and outputs from other subsystems.

---

7 Available at https://solcast.com/solar–radiation–data/
8 Available at http://www.fao.org/aquacrop/en/

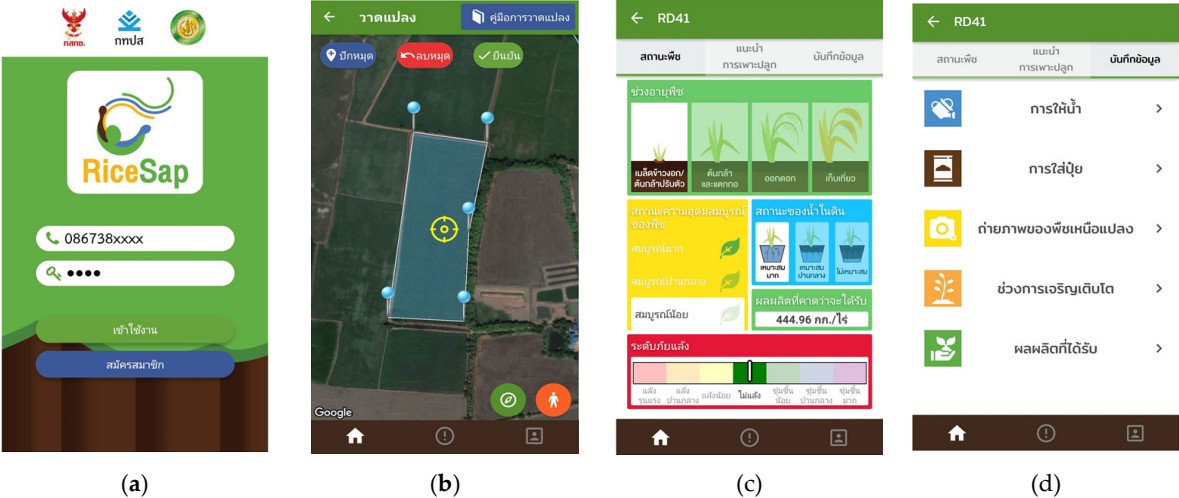

**Figure 15.** The RiceSAP mobile app: (**a**) welcome screen, (**b**) farmland registration, field practice, crop variety and planting date inputs; (**c**) crop monitoring functions, including rice stage, crop health, soil moisture, yield forecast and drought vulnerability index; (**d**) crop suggestion and management functions including watering and fertilizer schedules, taking crop photos, recording crop stages and actual yield.

## 5. Conclusions

Drought monitoring has been well developed in recent years. Unfortunately, drought impact assessment on crop yields has not shared the same trend due to the complex set of inputs required by crop simulation models. This work presents a methodology to combine both a drought monitoring model and a crop simulation model called SIMRIW−RS using satellite products. We show that satellite rainfall, LST, and NDVI products can be used for drought monitoring to generate a drought vulnerability index that can capture the dynamic of drought severity and duration in a given agricultural area. The same satellite rainfall and LST products are also proved to be effective weather inputs for SIMRIW−RS. When recalibrated with the Sentinel−2 LAI product, SIMRIW−RS can attain MAPEs of 5.12% and 3.54% for the 2018 and 2019 crop seasons, respectively. In addition, with 5-year historical satellite-derived weather datasets applied to SIMRIW−RS after recalibration, it can predict the rice yields approximately two months in advance with MAPEs of 5.08% and 5.02%. By observing time-series DVI values from a drought monitoring system, we can assess the effect of drought situations on the rice yield using SIMRIW−RS for a specified farmland. The result of the test site, based on the 2018 and 2019 crop season, exhibits consistent trends between time-series DVI and crop yields.

These promising results lead us to believe that, with big data in agriculture available from Thai government agencies together with machine learning techniques, SIMRIW–RS can be locally optimized for all regions of paddy fields. Since our approach is satellite-based, it can potentially be extended to operate nationwide, requiring very few inputs from farmers. Both drought monitoring and SIMRIW–RS models will become essential subsystems of a smart agriculture platform to be developed for crop monitoring and management. Once fully developed, the platform will be valuable to stakeholders in the rice supply chain, especially farmers and crop insurers to share the same information pertinent to a fair yield loss assessment. In addition, the ability to forecast rice yield two months in advance provides sufficient time for stakeholders to make informed decisions to carry out adaptation schemes for climate change.

**Author Contributions:** M.R. and W.V. performed the conceptualization, methodology, data analysis and preparation of the manuscript. K.H. and M.M. developed and maintained the SIMRIW−RS code and validated the results. K.O. administered field data collection, leading group discussions and conceptualization for the drought monitoring model. All authors have read and agreed to the published version of the manuscript.

**Funding:** This research was funded by the Science and Technology Research Partnership for Sustainable Development (SATREPS), JST–JICA, under the project "Advancing Co-design of Integrated Strategies with Adaptation to Climate Change in Thailand (ADAP–T)", sub-project "The Development of Drought Risk Analysis Platform using Multiple Satellite Sensors and Yield Estimation by AquaCrop/SIMRIW-RS Models with Satellite Drought Indices".

**Acknowledgments:** The authors acknowledge CSRS staff for their efforts in field data collection and handling the experiments. This work could not have been successful without their support.

**Conflicts of Interest:** The authors declare no conflict of interest.

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
