# Peer review of "Satellite-Based Drought Impact Assessment on Rice Yield in Thailand with SIMRIW−RS"

_remotesensing, doi:10.3390/rs12132099_

Round 1

Reviewer 1 Report

This study proposes a methodology that involves a combination of a drought monitoring model and crop simulation model using mostly space-based data to simulate rice yield. The test site for the study is a 6.88 ha paddy field. Overall, the study is interesting, but revisions are needed before it can be suitable for publication. Specific comments are provided below.

Line 20-21: Define all acronyms when first mentioned (i.e. LST, NDVI, etc.) in text. Also applicable to those mentioned in Line 94-95.

Line 23: Change “insufficient data availability” to “insufficient data”

Line 72: Change “Thailand which” to “Thailand, where”

Line 76: Please define and briefly describe NDVI here. Also, include a discussion of studies (cite studies) that demonstrate the use of soil moisture and NDVI as good indicators for drought and yield.

Line 104: This is the first time Sentinel LAI is mentioned- this should be introduced earlier, before stating it briefly here.

Line 97-106: The objectives are not clearly outlined. Please revised to clearly list the goal and/or objective(s) of this study.

Lines 110, 127, 136, 274, 288, 302, 312, 355, 366, 370, 391,403, 452: Something might be missing here - “Error! Reference source not found..”

Line 127-130: Add relevant citation/citations for definition of LAI

Line 131: Change to “Remote sensing”

Text in Figure 9 and Figure 10 are not clear

Methods: The methods section is not easy to follow; there are many subsections. I would suggest describing the overall methodology at the beginning of the section, then present all subsections. A flow chart at the beginning of the section would also be helpful so readers can clearly understand how different components are related as they read through the Methods.

Line 432: Though satellite products offer the regional coverage, it is important to note that the test site for the proposed methodology is only 6.88 ha (based on my understanding). Thus, are results on such a local scale enough to assume adoption at the regional scale? This should be discussed. Findings from other similar studies should be included and results compared.

Reviewer 2 Report

The figure references need correction through the whole manuscript. I suggest the authors resubmit the manuscript. Below is my detailed comments.

Figure 1: I am still confused about which one shows the study area boundary.
Figure 9 and 10 not readable
Line 240: How was the drought vulnerability index calculated?
Line 248: Legend please
Line 327: I suggest add a table shows inputs of SIMRIW–RS and selected data source.
Figure 6, 8, 16: legends are too small

Reviewer 3 Report

In this study, authors try to combine multiple data sources, including satellite images, data from weather stations as well as crop models to monitor drought and make a prediction of crop yield.

Although the sheer content of the work is very impressive, the presentation is unfortunately poor. I have extensive comments on the manuscript; I believe there are major changes that needs to be done in order to turn this manuscript into a more academic study rather than a governmental report. I'd like to suggest to authors to try to take a clear direction in the paper and try to focus on the path. This could be done by defining a clear research question/hypothesis and try to provide the only material and methods that is required to address the research question.

Round 2

Reviewer 1 Report

Authors addressed comments and suggested provided in the previous round of revisions, including clarifying objectives and added a methodology flowchart.I have a few additional, minor comments.

In addition to NDVI/LAI, and LST, authors must define all acronyms when first mentioned in text in text. For instance, what is GISTDA? All readers may not know these acronyms.

Line 86: SWRIW-RS is also mentioned in the abstract. Please clarify there as well.

Line 164-186: Why not move this section to the very beginning of the methods, immediately under “Materials and Methods”?

Line 167: change “details in” to “detailed in”

Reviewer 2 Report

The authors have addressed comments. The new version is much clearer to readers. I have a few additional comments.

  1. Please add more description about SIMRIW–RS, for example, inputs, outputs, relationship with DVI. None of the figures show the results of SIMRIW–RS, which is hard for readers to capture the concept.
  2. Please add more description about how do authors deal with the spatial resolution differences.
